# Enhanced Sensitivity to Subphonemic Segments in Dyslexia: A New Instance of Allophonic Perception

**DOI:** 10.3390/brainsci8040054

**Published:** 2018-03-26

**Authors:** Willy Serniclaes, M’ballo Seck

**Affiliations:** 1Speech Perception Lab., CNRS & Paris Descartes University, 75006 Paris, France; 2Human & Artificial Cognition Lab., Paris 8 University, 93526 Saint-Denis, France; m_ballo.seck04@univ-paris8.fr

**Keywords:** dyslexia, allophonic theory, speech perception

## Abstract

Although dyslexia can be individuated in many different ways, it has only three discernable sources: a visual deficit that affects the perception of letters, a phonological deficit that affects the perception of speech sounds, and an audio-visual deficit that disturbs the association of letters with speech sounds. However, the very nature of each of these core deficits remains debatable. The phonological deficit in dyslexia, which is generally attributed to a deficit of phonological awareness, might result from a specific mode of speech perception characterized by the use of allophonic (i.e., subphonemic) units. Here we will summarize the available evidence and present new data in support of the “allophonic theory” of dyslexia. Previous studies have shown that the dyslexia deficit in the categorical perception of phonemic features (e.g., the voicing contrast between /t/ and /d/) is due to the enhanced sensitivity to allophonic features (e.g., the difference between two variants of /d/). Another consequence of allophonic perception is that it should also give rise to an enhanced sensitivity to allophonic segments, such as those that take place within a consonant cluster. This latter prediction is validated by the data presented in this paper.

## 1. Introduction

Dyslexia is a specific deficit that renders an individual unable to acquire fluent reading skills in the absence of other cognitive deficits. Although dyslexia can be individuated in many different ways, it has only three discernable sources: a visual deficit that affects the perception of letters, a phonological deficit that affects the perception of speech sounds, and a bimodal (phono-visual) deficit that disturbs the association of letters with speech sounds [1] (Figure 1). Each of these three deficits is related to a different neural substrate and has a different genetic origin, suggesting that they can be present independently of each other in individuals with dyslexia [1]. Each deficit can in turn be expressed in different ways as it can possibly impact different cognitive dimensions such as perception, memory, and attention. These dimensions are interrelated [2] but each of them might be more or less severely affected in individuals with dyslexia depending on genetic and environmental factors.

Further, the exposure to written language will probably blur the distinction between the visual and auditory deficits. Visual confusions between letters will most probably destabilize the auditory representation of their phoneme counterparts; auditory confusions between phonemes will destabilize the visual perception of the corresponding letters; and the failure to establish letter-phoneme associations will affect both the visual and phonological representations. Such “migrations” between deficits will give rise to a vast array of individual profiles. On these grounds, individuation of the dyslexic profiles can be seen as the product of a highly specific substrate with an unsuccessful experience with the writing system. 

Given the wide individual variation in the manifestations of dyslexia and the restricted repertoire of deficits that are at its origin, one should try to specify as much as possible the very nature of each of these core deficits. Here we are interested in the phonological deficit of dyslexia that is generally attributed to a deficit in “phonological awareness”, that is, a deficit in the conscious access to phonemes and other phonological units [3]. While there is no doubt that a phonological awareness deficit is somehow related to dyslexia, an intriguing question is whether the phonological awareness deficit is at the origin of dyslexia or is instead the consequence of poor reading skills. Illiterate adults also exhibit a phonological awareness deficit, suggesting that written language deprivation might be a consequence, rather than a cause, of dyslexia [4]. However, early phonemic awareness, in pre-reading children, is a good predictor of future reading skills [5]. This difference between adults and children is probably related to the development of phoneme perception. The perception of speech with phonemic units results from a long-standing process from early childhood to the end of adolescence [6]. Becoming aware of the existence of phonemes, without the support of orthographic representations, is probably easier as long as the processes involved in phoneme perception are not fully automatized. In language learning, processes that take place below the level of subjective awareness are generally the case only for already established representations [7]. Illiterate adults have established phoneme representations, as evidenced by their capacity to perceive speech sounds in phoneme categories [8], which suggests that they have been aware of phonemes in the course of language development but lack the support of literacy to keep this faculty. 

A phonemic awareness deficit might thus be at the origin of dyslexia, but an important aim is to identify the very nature of this deficit. One possibility is that phonological awareness is only a matter of not having access to phonological units that are otherwise intact [9]. Another possibility is that people with dyslexia use units that are different from the phonemes and other phonological units. Here we will summarize the available evidence and present new data in support of this latter possibility.

### 1.1. Allophonic Theory

According to the “allophonic theory”, phonological dyslexia is due to a specific mode of speech perception that is characterized by the use of allophonic units, rather than phonemic ones [10]. Phonological representations are the end-product of a proacted developmental process. At the start, before some six months of age, the child is endowed with universal features that do not depend on language [11]. Features are differential units and correspond to qualitative changes in the value of some acoustic property (e.g., a change in the direction of a frequency transition) [12]. The universal features are “allophonic” in the sense that they can give rise to phonological features in some languages. Somewhat later, before one year of age, allophonic features are selected and combined to constitute “phonological” features that fit into the categories of the native language [11,13]. Phonological features are in turn combined into phonemic segments, a process that ends much later, not before the ages of five and six years old [6,14].

Phoneme perception would be fairly simple if the features corresponding to the same phoneme contrast were synchronized in the speech signal. However, features are dispersed over time in the signal, giving rise to acoustic segments without definite phoneme identity [15]. Such segments constitute potential phoneme segments in other languages and are “allophonic” by definition, that is, segments that are contextual variants of phonemes but do not pay independent contributions for distinguishing words in a given language (This definition of the allophone is similar to the one proposed by Trask (1996, p. 14); [16]: “One of two or more phonetically distinct segments which can realize a single phoneme in varying circumstances.”).

For example, in the French word /paRol/ (meaning ‘speech’), the segregation between /o/ and the surrounding /R/ and /l/ phonemes is based on changes in the frequencies of three different formants (F1, F2, and F3 “transitions”; Figure 2). However, these different acoustic changes are not synchronized and they do not coincide with the perceptual limits between /o/ and the surrounding phonemes. Such discrepancies give rise to different vocalic segments, a prototypical /o/ surrounded by /Ro/ and /ol/ transitional segments. These two latter segments are acoustically different from the prototypical /o/ and correspond to vowels that might constitute separate phonemes in some other languages.

Using allophonic units to perceive speech is possible but such units give a highly detailed description of speech sounds that is unnecessarily complex for accessing meaning. Writing systems were guided by a purpose of economy and they therefore used phonological units—at least this is what they did initially, although once printing was adopted writing did not always evolve with sound change. However, even transparent writing systems—those with one-to-one relationships between phonemes and graphemes—are eminently abstract for people who perceive speech with allophonic units, giving rise to major problems for reading acquisition. 

Allophonic theory claims that people affected by dyslexia do not integrate universal features into phonological ones during the development of speech perception [10]. Behavioral and neuro-physiological evidence in support of this theory has been collected in different studies. Discrimination responses to sounds varying along some acoustic continuum (a F2–F3 continuum for place-of-articulation distinctions; a VOT (“Voice Onset Time” [18]) continuum for voicing distinctions) show that individuals with dyslexia are sensitive both to allophonic and phonemic boundaries along the continuum, whereas typical-reading controls are only sensitive to phonemic boundaries [10,19,20,21,22]. For this reason, allophonic perception is indeed associated with dyslexia, but some degree of phonemic perception is also present though to a smaller extent than the one evidenced in typical readers. Further, children with dyslexia do not always exhibit an enhanced discrimination of allophonic boundaries (e.g., [23]), suggesting that these children would not use allophonic units. However, other studies showed that even when behavioral data did not show a sensitivity to allophonic boundaries in dyslexia, it was nevertheless present in neural recordings (in children at risk of dyslexia: [21]; in adults with dyslexia: [22]). In summary, dyslexia does not exclude the acquisition of some degree of phonemic perception but it remains in competition with allophonic perception even though the latter does not always emerge at the behavioral level.

The question then is why should such competition still raise severe problems for learning to read? The persistence of reading problems in the children who develop dyslexia might arise from the fact that their access to phonemic representations is delayed by the pop out of allophonic representations. Studies on the bimodal perception of letters and speech sounds have clearly demonstrated that the synchronization of the visual and auditory inputs has crucial importance for learning to read [24].

### 1.2. The Present Study

The previous evidence in support of allophonic perception in dyslexia was based on an enhanced perception of allophonic features. However, allophonic perception should also give rise to an enhanced perception of allophonic segments. Normally, the acoustic transitions between phonemes participate in the identification of the surrounding phonemes [25] but they do not give rise to perception of separate segments. Things might turn out differently for listeners with dyslexia whenever transitional speech segments correspond to some allophonic category. For example, the transitions inside a consonant cluster like /Rl/ might contain a vocalic segment that corresponds to a vowel in some other language, and people with dyslexia might then use their allophonic sensitivity to perceive such a vowel. This is the hypothesis that motivated the present study. 

In order to test this hypothesis, the duration of the vocalic segment inside a /Rl/ cluster in the French word /paRol/ was progressively reduced, eventually giving rise to the perception of the word /paRl/ (meaning ‘speak’). Following allophonic theory, the paRol/paRl boundary (i.e., the 50% response point on the durational continuum) should be located at a shorter /o/ duration for children with dyslexia, compared to typical-reading children, because they attribute a separate identity to the transitional segments between phonemes, e.g., between /R/ and /l/. The /^R^o^l^/ segment that was temporally reduced to create the paRol/paRl continuum is composed of a prototypical /o/ (in its central portion) surrounded by /Ro/ and /ol/ transitional segments. These two latter segments are acoustically different from the prototypical /o/ and correspond to vowels that might constitute separate phonemes in some other languages. Allophonic theory postulates that people with dyslexia perceive such ‘allophones’ as separate vowel units, not as transitions between /R/ and /l/. Accordingly, people with dyslexia should rely on these vowel allophones to perceive a vowel inside the /Rl/ cluster at relatively short /^R^o^l^/ durations.

## 2. Materials and Methods

Stimuli: The word /paRol/, pronounced by a male French speaker, was used to generate a paRol/paRl continuum by progressive temporal reduction of the /o/ segment with Soundforge© [26]. The /o/ limits were identified by ear by two trained phoneticians (the authors). The starting point corresponded to the end of the /R/ segment and the endpoint to the beginning of a clear /l/ segment, both without discernable trace of a vocalic influence. The segment between these limits included formant transitions from /R/ to /o/ as well as those from /o/ to /l/ (Figure 2). To remember that this segment contained such transitions we will label it /^R^o^l^/. The /^R^o^l^/ segment was progressively reduced from 126 ms to 9 ms by erasing segments of about 15 ms that were centered on the middle of the /^R^o^l^/ segment, generating nine different stimuli.

Task: Each of the nine stimuli was presented 10 times for identification. The stimuli were presented in a random order with a 1000 ms intertest interval. A Matlab^©^ [27] program (Version 6.1) was used to deliver the stimuli and collect the responses. The stimuli were delivered by Samson Professional Studio^©^ headphones in a quiet room. The responses were given by pressing a dedicated key on the keyboard as rapidly as possible. The participants were first familiarized with the task using the continuum endpoints. The same procedure was used for the children and the adults. The whole procedure took about 15 to 20 min.

Participants: Nineteen French-speaking children with dyslexia (7 females; hereafter “dyslexic” DYS children) were recruited with the help of two speech therapists working in the Paris area. The children had no oral language disorder (as reported by their therapists) and showed a reading delay larger than 2 years (95% C.I. limits: −2.3, −3.4 years) on the Alouette Reading Test [28], which is based on the precision and speed in the reading of a short text (279 words). 

Fifty-two typically developing French-speaking children (25 females; hereafter “control” (CTL) children) were recruited from a school in the Paris urban area. They reported no history of oral language or reading disorder and showed a normal reading age (95% C.I. limits: 0.73, 1.38 years) on the Alouette Reading Test (see Table 1). 

Twenty-six typical French-speaking adults (13 females), who reported no history of oral language or reading disorder, were also recruited (mean age; SD: 26.3; 3.75 years; hereafter “control” (CTL) adults).

Analysis strategy: A fairly large sample of 52 typical children (25 females) was taken in order to get a subsample of the same reading age as the DYS group (19 children; 7 females). However, in order to assess the difference between CTL and DYS on the largest possible basis we first draw a comparison between the DYS and the whole CTL group. These groups differed in both reading age and chronological age (Table 1). However, the confounding effect of reading age was controlled by comparing the results of the DYS group with those of the subgroup of reading age controls (Table 2). The confounding effect of chronological age was controlled by comparing the results of the whole CTL group and a group of typical adults. 

The characteristics of the subsamples of children matched for reading age are presented in Table 2. The 12 DYS children (3 females) had much the same reading as the 12 CTL children (4 females), but they were older and exhibit a larger reading delay than the CTL. The reading delay was only significantly different from zero for the DYS children (95% C.I. limits: −2.7, −3.9 years; 0.03, −0.46 years; for the DYS and CTL, respectively).

Data analysis: The boundary and the slope of the identification functions were assessed for each subject using Logistic Regression (SPSS 24^©^ [29], version 24.0), with the identification response as the dependent variable and the stimulus as the independent variable. Discriminant Analysis (SPSS 24^©^), with drop-one-out cross-validation method was used to assess the reliability of the perceptual differences across individuals.

## 3. Results

### 3.1. Whole Samples

Figure 3 shows that the paRol/paRl boundary was located at a longer /^R^o^l^/ duration (65 ms location; SD = 5.1 ms) for the CTL children than for the DYS children (50 ms location; SD = 5.4 ms). The difference in boundary location between the two groups of children was significant (F(1,69) = 116; *p* < 0.001; η^2^ = 0.627). This difference in boundary location was fairly small (about 15 ms) but it was highly reliable. When the boundary location was used to reclassify the children as either DYS or CTL with Discriminant Analysis, there was about 90% correct reclassification (cross-validated classification with one-drop-out). 

There also was a slight (but significant) difference in the slope of the identification function between the two groups (F(1,69) = 4.70; *p* < 0.05; η^2^ = 0.064). The slope was shallower for the DYS compared to the CTL (Figure 3).

### 3.2. Controlling the Effect of Chronological Age

There was a significant difference in the location of the paRol/paRl boundary between the CTL children and adults (F(1,76) = 35.3, *p* < 0.001, η^2^ = 0.317). The adults’ boundary was located at a *longer* duration than the children’s one (72 ms, SD = 3.2; 65 ms, SD = 5.4; respectively; see Figure 3), showing a decrease with age. As the DYS children were older than the CTL children (whole samples, Table 1), the fact that their boundary was located at a shorter duration than that of the CTL children can hence not be readily explained by the difference in chronological age between the two groups. However, a nonlinear effect of age cannot be entirely ruled out. 

There also was a slight (but significant) difference in the slope of the identification function between the two groups (F(1,76) = 5.29; *p* < 0.05; η^2^ = 0.065). The slope was shallower for the children compared to the adults (Figure 3).

### 3.3. Controlling the Effect of Reading Age

Although the difference in boundary location between the DYS and CTL children is most probably due to a difference in reading status between the two groups, not to the difference in chronological age, the effect of the reading status might be mediated by a difference in reading age. In order to control a possible confounding effect of reading age, subsamples of DYS and CTL children with the same reading age (Table 2) were compared. 

Figure 4 gives the identification functions of the DYS and reading level CTL (matched samples, Table 2). The difference in the location of the identification boundary between the two groups was again highly significant (F(1,22) = 32.8, *p* < 0.001, η^2^ = 0.599). As expected, the DYS identification boundary was located at a shorter duration (50 ms; SD = 5.5) for the DYS than for the reading CTL (64 ms; SD = 6.8). 

As for the full samples of children, the difference in boundary location was fairly small (about 14 ms) but it was again highly reliable. When the boundary location was used to reclassify the children as either DYS or CTL with Discriminant Analysis, there was about 88% correct reclassification. 

The similarities between the results obtained with the matched subsamples with those obtained with the whole samples indicate that the difference in boundary location between the DYS and the CTL children, which was evidenced between the two full samples, was not due to the difference in reading age between the two groups. Dyslexia per se, rather than the related difference in reading age, was at the origin of the perceptual difference between the groups. 

The slope of the identification function did not significantly differ between the DYS and the reading age controls, although there was a trend (F(1,22) = 3.03, *p* = 0.10, η^2^ = 0.121).

## 4. Discussion

The results of this study evidenced a difference in the perception of a paRol/paRl contrast between children with dyslexia and typically reading children. The paRol/paRl boundary was located at shorter /o/ durations for the children with dyslexia compared to the typically reading ones. 

Such perceptual differences did not seem to be related to the difference in chronological age between the DYS and CTL children in this study. The comparison between CTL children and adults suggests that the effect of age is to shift the boundary towards a longer duration. If age was responsible for the difference in boundary location between the DYS and the CTL children in this study, the paRol/paRl boundary should have been located at a longer /o/ duration for the DYS children. Although a nonlinear effect of age remains possible, this would mean that the boundary would decrease between the ages of seven and nine years old (the mean ages of the CTL and DYS children in the present study) before rising between nine years and adulthood. While a nonlinear effect of age on allophonic perception has indeed been reported in the literature [30], the inflection point was located around seven years old and is therefore not compatible with the inflection at nine years old that would underlie a non-linear effect in the present study. Finally, the difference in boundary location between the seven year-old children and the adults might be due to non-perceptual factors that covary with age. However, non-perceptual factors might also be at the origin of the boundary difference between the seven and nine year-old children. Whatever its origin, an effect of age on the location of the boundary should go in the same direction for both comparisons (i.e., those between children and between children and adults). 

The DYS children had a younger reading age than the CTL children (Table 1). However, a confounding effect of reading experience can be safely ruled out on the grounds that the difference in boundary location between the full samples of DYS and CTL children was replicated by a quite similar difference between subsamples of DYS and CTL children of the same reading age. 

The difference in boundary duration between the DYS and CTL children was highly consistent. Boundary location was able to assign the children as having dyslexia or not with 90% correct classification, a score that is fairly large in absolute terms. Such a score is also larger than comparable values reported in the literature. The location of the boundary on a do/to continuum allowed 81% correct classification in children with dyslexia and chronological age controls, and 70% correct classification in children with dyslexia and reading level controls [19].

### Theoretical Implications

These results are impressive for two reasons. Firstly, the relationship between dyslexia and the perception of a paRol/paRl contrast was entirely based on theoretical grounds. The results followed a prediction of the allophonic theory without any prior empirical evidence. Remember that following allophonic theory, the paRol/paRl boundary should be located at a shorter /o/ duration for people with dyslexia because they attribute a separate identity to the transitional segments between /R/ and /l/, and that they should rely on these allophonic segments to perceive a vowel inside the /Rl/ cluster at relatively short /^R^o^l^/ durations, a prediction that was verified in the present results.

Secondly, the high degree of reliability of the difference in boundary location along the parol/parl continuum suggests that it is closely related to the core difference in phonemic perception between children with dyslexia and typical-reading children.

Allophonic theory readily accounts for the enhanced sensitivity of DYS children to vocalic segments inside a consonant cluster. However, another possible explanation is that people with dyslexia perceive the acoustic signal with a higher temporal acuity. According to the Temporal Sampling Framework (TSF), brain generators oscillating at higher rates would be at the origin of a large set of deficits evidenced in dyslexia, including allophonic perception [31,32]. The atypical neural responses of the DYS to relative high frequency modulations (in the beta-gamma range) would have as direct consequence in that they perceive “subphonemes” (i.e., allophonic segments) [33]. If higher temporal acuity was at the origin of allophonic perception, the enhanced perception of short speech segments by the DYS should not depend on their spectral content. DYS should also be more sensitive to a spectrally uniform vowel inside a consonant cluster, but this remains to be proved (Preliminary data suggest that there is no difference in boundary location between children with dyslexia and chronological age control children along a paRol/paRl continuum that was generated by temporal reduction of a /o/ segment with constant formant frequencies (i.e., the /^R^o^l^/ segment was replaced by a segment generated by reduplication of a central period at the middle of the segment). 

The enhanced sensitivity of DYS children to allophonic segments fits pretty well with the results of previous studies that evidenced an enhanced sensitivity to allophonic features in children or adults with dyslexia, in at least three different languages (French, Dutch, Spanish; see Introduction). Overall, taking account of the present results on segmental allophony with the previous results on contrastive allophony, a majority of the children affected with dyslexia display allophonic perception. These are probably those who also display a deficit in phoneme awareness and are classically considered as affected by “phonological” dyslexia. One pending question however is whether allophonic perception is directly responsible for their specific failure in reading acquisition. 

A recurring question in the recent literature concerns the implications of allophonic perception for learning to read. On purely deductive grounds, perceiving more sounds than there are letters to represent them (featural allophony) and perceiving extra sounds between two successive letters (segmental allophony) should raise serious difficulties in capturing the regularities that govern a writing system. In a study with English-speaking school-age children, brain event-related potentials to a Finnish phoneme contrast were better correlated to reading skills than those with an English phoneme contrast; the higher the correlation with the foreign contrast the poorer the reading skills [34]. However, there are hints that some children might overcome these difficulties while others might not. The effect of a Categorical Perception deficit (CP deficit, a proxy for allophonic perception, e.g., [35]) on reading was mediated by a deficit in phonemic awareness in a study with French children of about 10 years old with dyslexia [36]. However, there was no evidence for a cascading effect from a CP deficit to reading fluency in a study with Dutch children of about 9 years old at familial risk for dyslexia [37]. For these children, the relationship between the CP deficit and reading was mediated by RAN (Rapid Automatized Naming). A possible interpretation of both the French and Dutch results is that the consequences of allophonic perception for reading depend on other factors, and that the latter vary with the writing system. Remember that the Dutch writing system is more transparent that the French one [38], and that specific reading difficulties are mainly a matter of reading fluency in languages with a fairly transparent orthography, whereas they are also a matter of reading accuracy in languages with a more opaque orthography [39]. Consequently, the effect of allophonic perception on reading might be modulated either by timing constraints, explaining the mediation by RAN in Dutch, and also by the access to phoneme representations, explaining the mediation by phonemic awareness in French. 

Timing constraints might impact the synchronization of grapheme and phoneme decoding in the temporal cortex, which is vital for learning to read [24]. Phoneme decoding is slower than allophonic decoding in children with dyslexia [21], a difference that might be larger in children with a RAN deficit. The temporal delay between phonemic and allophonic decoding might also vary with individual differences in the access to phonological representations in the prefrontal cortex, such differences being correlated with phonemic awareness [9]. A higher degree of phonemic awareness might correspond to a quicker access to the prefrontal cortex and, in turn, a faster transmission of the phonemic codes to the temporal cortex. Finally, it should be stressed that it is not only the access to the prefrontal cortex but also the nature of the prefrontal representations (phonemic vs. allophonic: [40]) that might affect the transmission of the phonemic codes to the temporal cortex.

## 5. Conclusions

The present results showed that children with dyslexia exhibited an increased sensitivity to allophonic segments, in accordance with previous results that evidenced an increased sensitivity to allophonic features. While it seems pretty clear that allophonic representations should raise severe obstacles for reading acquisition, their actual consequences are probably mitigated by the competition with phonemic representations in the brainsi8 of the children with dyslexia. The implications of allophonic perception for reading acquisition are modulated by factors such as RAN and phonemic awareness skills that might reflect individual differences in the synchronization of grapheme and phoneme decoding. Knowing more precisely how these factors regulate such timing might help to better understand (and remediate) dyslexia. 

## Figures and Tables

**Figure 1 brainsci-08-00054-f001:**
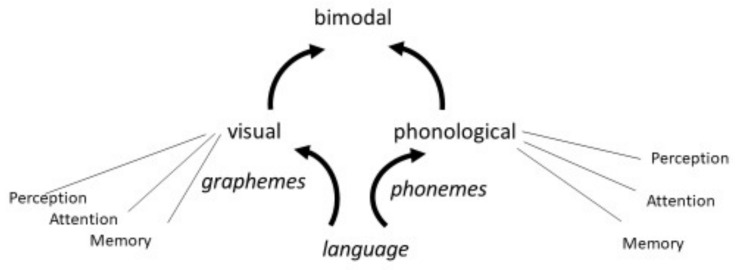
The three sources of dyslexia: phonological, visual, and bimodal deficits.

**Figure 2 brainsci-08-00054-f002:**
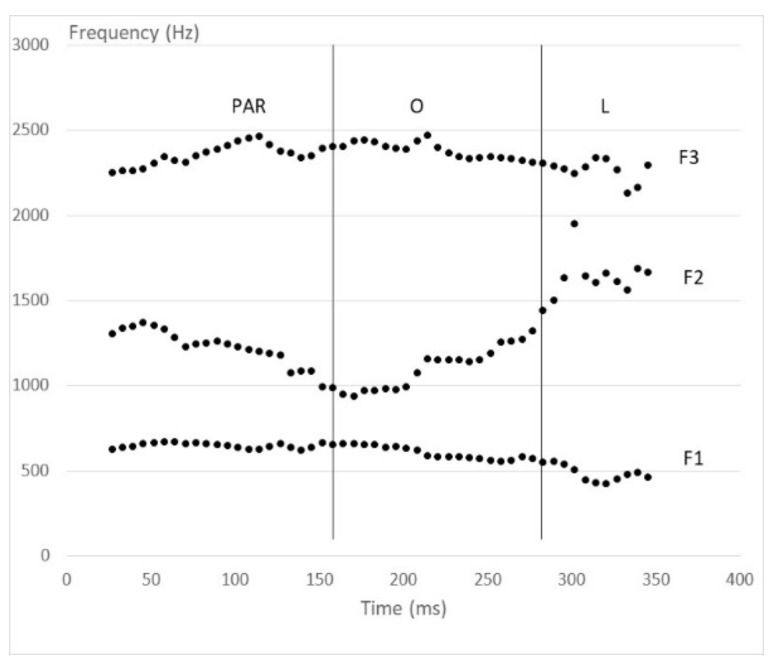
Formant frequencies (Praat^©^ [17], version 6.0.37) in the word /paRol/ produced by a French speaker. The vertical lines correspond to the perceptual limits between the /^R^o^l^/ segment and the initial (PAR) and final (L) parts of the word. F1, F2, F3 correspond to formants one, two and three, respectively.

**Figure 3 brainsci-08-00054-f003:**
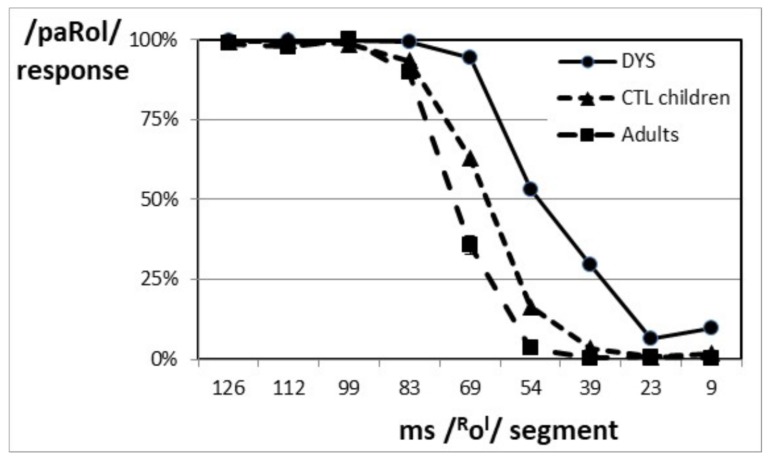
Identification functions of the whole samples of DYS and CTL children, and of CTL adults.

**Figure 4 brainsci-08-00054-f004:**
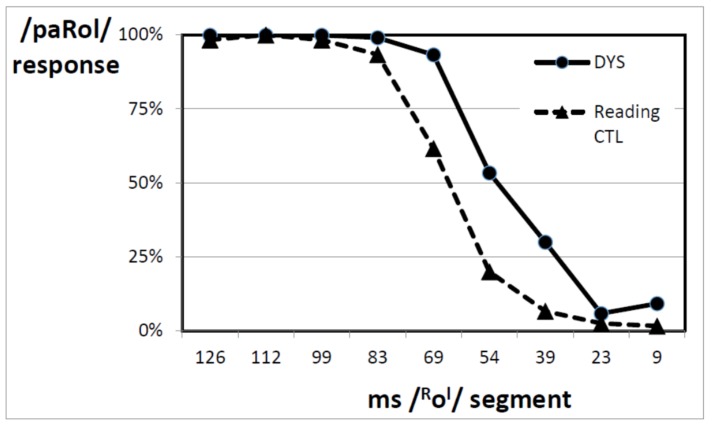
Identification functions of subsamples of DYS and CTL children with the same reading age.

**Table 1 brainsci-08-00054-t001:** Characteristics of the full samples of children with dyslexia (“DYS”) and control children (“CTL”).

Full Samples	Chronological Age Mean (SD)	Reading Age Mean (SD)	Reading Delay Mean (SD)
CTL (*N* = 52)	7.60 years (0.36)	8.65 years (1.23)	+1.06 years (1.18)
DYS (*N* = 19)	9.55 years (1.51)	6.68 years (0.61)	−2.87 years (1.15)
Difference	F(1,69) = 77.4 (*p* < 0.001)	F(1,69) = 43.9 (*p* < 0.001)	F(1,69) = 155 (*p* < 0.001)

**Table 2 brainsci-08-00054-t002:** Characteristics of subsamples of children with dyslexia (“DYS”) and control children of the same reading age (“Reading Age CTL”).

Matched Samples	Chronological Age Mean (SD)	Reading Age Mean (SD)	Reading Delay Mean (SD)
Reading Age CTL (*N* = 12)	7.42 years (0.29)	7.20 years (0.29)	−0.22 years (0.39)
DYS (*N* = 12)	10.3 years (1.11)	7.02 years (0.51)	−3.31 years (0.96)
Difference	F(1,22) = 77.0 (*p* < 0.001)	F(1,22) = 1.12 (*p* = 0.30)	F(1,22) = 107 (*p* < 0.001)

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
