# Peer review of "Enhanced Sensitivity to Subphonemic Segments in Dyslexia: A New Instance of Allophonic Perception"

_brainsci, 2018, doi:10.3390/brainsci8040054_

Reviewer 1 Report

The manuscript “Enhanced sensitivity to subphonemic segments in dyslexia: A new instance of allophonic perception” reports the findings of an experiment that investigated the ability to identify the contrast between two words based on an allophonic segment boundary in French. The participants were school-aged children with and without dyslexia and a group of adults with typical reading skills. Results showed that children with dyslexia interpreted a shorter boundary location as indication of the contrast than controls, and this was true when they were compared to all controls involved in the study and a smaller sub-group of reading-age matched controls. Adults required a longer boundary than the two groups of children.

This study focuses on the perception of allophonic segments, which sets it apart from previous research investigating speech perception in dyslexia. The findings align with the allophonic theory’s prediction that individuals with dyslexia perceive non-phonemic contrasts in the speech input including contrasts between individual phoneme contrasts and, as shown here, between segments. I believe that for this reason, this study makes an important contribution to the current literature on dyslexia, but I have a number of concerns that could be addressed in a revision to the manuscript.

-       I appreciate the authors’ effort to keep the introduction and literature review concise, but I think that the paper would benefit from a more detailed discussion of the differences between a phonological awareness deficit in dyslexia and a phonological perception deficit. The authors mention that it is possible that “phonological awareness is only a matter of not having access to phonological units that are otherwise intact” or that “people with dyslexia use units that are different from the phonemes and other phonological units”. While phonological perception and phonological awareness are related, it would be clearer if the authors explained how the two concepts are different and the evidence that phonological awareness may have other causes (e.g., it may actually stem from the reading difficulties that children have).

-       Lines 51-53: ‘lack of phonological awareness’ – I think that the authors are referring to a deficit in phonological awareness rather than a complete lack of it.

-       Lines 90-94:  this is related to my first comment as it appears that the terms phonemic awareness and phonemic perception are used interchangeably here. These two sentences can be revised to improve clarity.

-       I was not able to see a procedures section in the Method or a description of the task. I assume that it was an identification task typically used in this type of studies, but it would be helpful if this were clarified in the paper. It is also unclear whether an identical task was used for children and adults.

-       My main concern is with the selection of participants for the study and the justification provided for the inclusion of the three groups. I agree that including adults in this study is useful since this task involves stimuli that have not been used before, so the adults provide an indication of the mature performance in this task. However, I do not believe that the claim that the inclusion of adults provides a test for the effects of chronological age on task performance is justifiable. Adults and children differ on many more dimensions than age that could also affect their performance in the task (attention, memory, etc.).

-       On a related note, the explanation for the selection of the two groups of children is not clear. The complete sample included children with and without dyslexia, whereby the controls differed from dyslexics on chronological age, lexical age, and reading age (according to Table 1). Therefore, it is not clear how these two groups of children are comparable and why a group of chronological age-matched controls was not included.

-       In the first column of Tables 1 and 2, the CTL group is described as ‘reading age’. Is this an error in Table 1 since the entire sample was not matched on chronological or reading age?

-       Tables 1 and 2 also report lexical age for all children. It is not clear how these results were obtained. Are they also calculated based on children’s performance on the Alouette Reading test? If so, how do they differ from reading age?  

-       Section 3.3 appears to conflate the terms reading age and reading experience. Since the children in the control group were significantly younger than the children in the dyslexia group, it can be argued that they actually had less reading experience (as they would have spent less time in school and received less instruction). For this reason, I would suggest that the authors use the term reading age consistently as this is what is measured in this study.

-       Minor points:

o   Line 195 – typo “adult’s boundary of was”

o   Is it possible to use a different marker for the line showing adult performance on Figure 3? At the moment both CTL and adult lines use triangles, which makes it difficult to differentiate between the two lines (this may not be an issue if the figure is larger in the final version of the manuscript). 

Author Response

ANSWERS TO THE REVIEWER are inserted below

We also added an argument in support of the relationship between allophonic perception and reading skills:

In a study with English-speaking school-age children, brain event-related potentials to a Finnish phoneme contrast were better correlated to reading skills than those with an English phoneme contrast, the higher the correlation with the foreign contrast the poorer the reading skills [31].

31.   Hämäläinen, J.; Landi, N.; Loberg, O.; Lohvansuu, K. ; Pugh, K.; Leppänen, P.H.T. Brain event-related potentials to phoneme contrasts and their correlation to reading skills in school-age children. Int. J. Behav. Devpt. 2017,

             https://doi.org/10.1177/0165025417728582  DOI: 10.1177/0165025417728582

OpenReview

English language and style

( ) Extensive editing of English language and style required
( ) Moderate English changes required
(x) English language and style are fine/minor spell check required
( ) I don't feel qualified to judge about the English language and style

Yes

Can be   improved

Must be   improved

Not   applicable

Does the introduction provide sufficient background and include all   relevant references?

( )

(x)

( )

( )

Is the research design appropriate?

( )

( )

(x)

( )

Are the methods adequately described?

( )

( )

(x)

( )

Are the results clearly presented?

( )

(x)

( )

( )

Are the conclusions supported by the results?

( )

(x)

( )

( )

Comments and Suggestions for Authors

The manuscript “Enhanced sensitivity to subphonemic segments in dyslexia: A new instance of allophonic perception” reports the findings of an experiment that investigated the ability to identify the contrast between two words based on an allophonic segment boundary in French. The participants were school-aged children with and without dyslexia and a group of adults with typical reading skills. Results showed that children with dyslexia interpreted a shorter boundary location as indication of the contrast than controls, and this was true when they were compared to all controls involved in the study and a smaller sub-group of reading-age matched controls. Adults required a longer boundary than the two groups of children.

This study focuses on the perception of allophonic segments, which sets it apart from previous research investigating speech perception in dyslexia. The findings align with the allophonic theory’s prediction that individuals with dyslexia perceive non-phonemic contrasts in the speech input including contrasts between individual phoneme contrasts and, as shown here, between segments. I believe that for this reason, this study makes an important contribution to the current literature on dyslexia, but I have a number of concerns that could be addressed in a revision to the manuscript.

-       I appreciate the authors’ effort to keep the introduction and literature review concise, but I think that the paper would benefit from a more detailed discussion of the differences between a phonological awareness deficit in dyslexia and a phonological perception deficit. The authors mention that it is possible that “phonological awareness is only a matter of not having access to phonological units that are otherwise intact” or that “people with dyslexia use units that are different from the phonemes and other phonological units”. While phonological perception and phonological awareness are related, it would be clearer if the authors explained how the two concepts are different and the evidence that phonological awareness may have other causes (e.g., it may actually stem from the reading difficulties that children have).

ANSWER: We tried to conciliate the results on illiterate people with those of pre-readers.

ADDED p. 3, lines 56-73:While there is no doubt that a phonological awareness deficit is somehow related to dyslexia, an intriguing question is whether the phonological awareness deficit is at the origin of dyslexia or is instead the consequence of poor reading skills. Illiterate adults also exhibit a phonological awareness deficit, suggesting that written language deprivation might be a consequence, rather than a cause, of dyslexia [4]. However, early phonemic awareness, in pre-reading children, is a good predictor of future reading skills [5]. Such difference between adults and children is probably related to the development of phoneme perception. The perception of speech with phonemic units results from a long-standing process from early childhood to the end of adolescence [6]. Becoming aware of the existence of phonemes, without the support of orthographic representations, is probably be easier as long as the processes involved in phoneme perception are not fully automatized. In language learning, processes that take place below the level of subjective awareness are generally the case only for already established representations [7]. Illiterate adults have established phoneme representations, as evidenced by their capacity to perceive speech sounds in phoneme categories [8], which suggests that they have been aware of phonemes in the course of language development but lack the support of literacy to keep this faculty.

-       Lines 51-53: ‘lack of phonological awareness’ – I think that the authors are referring to a deficit in phonological awareness rather than a complete lack of it.

MODIFIED

-       Lines 90-94:  this is related to my first comment as it appears that the terms phonemic awareness and phonemic perception are used interchangeably here. These two sentences can be revised to improve clarity.

MODIFIED lines 126-130

“For this reason, allophonic perception is indeed associated with dyslexia, but some degree of phonemic perception is also present but to a smaller extent than the one evidenced in typical readers. Further, children with dyslexia do not always exhibit an enhanced discrimination of allophonic boundaries (e.g. [22]), suggesting that these children would not use allophonic units.

-       I was not able to see a procedures section in the Method or a description of the task. I assume that it was an identification task typically used in this type of studies, but it would be helpful if this were clarified in the paper. It is also unclear whether an identical task was used for children and adults.

WE APOLOGIZE FOR SUCH OMISSION

ADDED lines 184-190

“Task. Each of the nine stimuli was presented 10 times for identification. The stimuli were presented in a random order with a 1000 ms intertest interval. A MatlabÓ program was used to deliver the stimuli and collect the responses. The stimuli were delivered by Samson Professional StudioÓ headphones in a quiet room. The responses were given by pressing a dedicated key on the keyboard as rapidly as possible. The participants were first familiarized with the task using the continuum endpoints. The same procedure was used for the children and the adults. The whole procedure took about 15 to 20 minutes.

-       My main concern is with the selection of participants for the study and the justification provided for the inclusion of the three groups. I agree that including adults in this study is useful since this task involves stimuli that have not been used before, so the adults provide an indication of the mature performance in this task. However, I do not believe that the claim that the inclusion of adults provides a test for the effects of chronological age on task performance is justifiable. Adults and children differ on many more dimensions than age that could also affect their performance in the task (attention, memory, etc.).

ANSWER : WE KEPT THE ADULT GROUP FOR REASONS THAT ARE NOW GIVEN IN THE METHOD AND THE DISCUSSION.

MODIFIED Method Lines 208-215

“Analysis strategy. A fairly large sample of typical children (# 52) was taken in order to get a subsample of the same reading age as the DYS group (# 19). However, in order to assess the difference between CTL and DYS on the largest possible basis we first draw a comparison between the DYS and the whole CTL group. These groups differed in both reading age and chronological age (Table 1). However, the confounding effect of reading age was ruled out by comparing the results of the DYS group with those of the subgroup of reading age controls (Table 2). The confounding effect of chronological age was ruled out by comparing the results of the whole CTL group and a group of typical adults.”

MODIFIED Discussion Lines 306-311

“Finally, the difference in boundary location between the 7 yr. old children and the adults might be due to non-perceptual factors that covary with age. However, non-perceptual factors might also be at the origin of the boundary difference between the 7 and 9 yrs. old children. Whatever its origin, an effect of age on the location of the boundary should go into the same direction for both comparisons (i.e. those between children and between children and adults).

-       On a related note, the explanation for the selection of the two groups of children is not clear. The complete sample included children with and without dyslexia, whereby the controls differed from dyslexics on chronological age, lexical age, and reading age (according to Table 1). Therefore, it is not clear how these two groups of children are comparable and why a group of chronological age-matched controls was not included.

ANSWER:  NOW EXPLAINED IN THE METHOD (SEE PREVIOUS POINT).

-       In the first column of Tables 1 and 2, the CTL group is described as ‘reading age’. Is this an error in Table 1 since the entire sample was not matched on chronological or reading age?

YES IT WAS AN ERROR. WE APOLOGIZE. MODIFIED

-       Tables 1 and 2 also report lexical age for all children. It is not clear how these results were obtained. Are they also calculated based on children’s performance on the Alouette Reading test? If so, how do they differ from reading age? 

MODIFIED: LEXICAL AGE NOW REPLACED BY READING AGE

-       Section 3.3 appears to conflate the terms reading age and reading experience. Since the children in the control group were significantly younger than the children in the dyslexia group, it can be argued that they actually had less reading experience (as they would have spent less time in school and received less instruction). For this reason, I would suggest that the authors use the term reading age consistently as this is what is measured in this study.

MODIFIED ACCORDINGLY.

-       Minor points:o   Line 195 – typo “adult’s boundary of was”

CORRECTED

o   Is it possible to use a different marker for the line showing adult performance on Figure 3? At the moment both CTL and adult lines use triangles, which makes it difficult to differentiate between the two lines (this may not be an issue if the figure is larger in the final version of the manuscript). 

MODIFIED

Reviewer 2 Report

Summary

Thank you for the opportunity to review the manuscript “Enhanced sensitivity to subphonemic segments in dyslexia: a new instance of allophonic perception”. This manuscript describes an interesting experiment about subtle differences in speech perception between French typically reading children and adults, and children with dyslexia. As a consequence of co-articulation (where articulation of one sound is influenced by its surrounding sounds), the /o/ in /paRol/ (French for ‘speaking’) is influenced by the /R/ and the /l/. Taking the ‘allophonic mode of speech perception’ theory as a starting point, the authors expected that participants with dyslexia might not perceive the parts of the vowel that includes transitions from the /R/ and the /l/ as belonging to the vowel, but as separate vowels. This would lead to a different location for the phonemic boundary between /paRol/ and /paRl/ for participants with dyslexia. That was indeed what was found.

Broad comments

This is an interesting study, using a clever experimental manipulation to test the ‘allophonic mode of speech perception’ in a new way. The design seems sound and the interpretation of the results seems warranted, but I do a several broader and a rather long list om more specific comments.

- I found the manuscript a bit hard to follow, because of the large number of terms used to refer to phonological and allophonic aspects of speech perception, without explaining the differences between these terms. What for instance do the authors mean by phonological features and phonemic segments, and what are phonemic activations? Similarly, a clearer definition of allophonic units vs. allophonic features vs. allophonic segments vs. allophonic categories vs. allophonic activations should be given. And what are subphonemes (p12, l290)?

- A more detailed explanation of the relation between the ‘allophonic theory’ and the experimental design early on would make it easier to follow the rest of the paper. In the discussion (p11, l272-282) the authors provide such an explanation. I’d suggest moving that section to the introduction.

- An more detailed explanation of the speech perception task (e.g., how many trials were included for each step along the continuum?, where trials randomized?, how long was the task?) and the procedure (where were the participants tested, how long did it take?, how was made sure that the task was engaging for children?) is essential in order for the reader to fully appreciate the experiment.

- On p11, l262+l263 (+elsewhere): “people/adults/children with dyslexia” would be more appropriate than “dyslexics”.

Specific comments

Introduction

- In the first line of the introduction (p1, l28) the authors state that “Dyslexia is a specific deficit that renders an individual unable to learn to read…”. That seems a bit too drastic as most individuals with dyslexia do learn to read, but reading remains inaccurate, slow and effortful. A more nuanced definition of dyslexia would be appropriate.

- On p2, l35, after “dyslexia.” a reference should be included.

- On p2, l35-36: the statement “Each deficit can in turn be expressed in different ways depending on its impact on perception, memory and attention.” is a little vague. At the very least references should be included to make clearer what the authors mean.

- On p2, l36-37: it is unclear what the authors mean by “interact in various different ways”. Clarify, or if the next sentence is meant as clarification, please make that connection more explicit.

- On p2, l38: “audiological” should probably be “auditory”.

- On p2, l44, after “writing system.” a reference should be included.

Materials and Methods

- On p5, l128: after “Soundforge” a reference should be included.

- On p6, l143: I don’t know what “orthophonist practitioners” are.

- On p6, l144: The authors mention that the children had no oral language disorder. It should be explained on the basis of which assessment that was concluded.

- On p6, l145: adding a brief explanation of the “Alouette Reading Test” would be appropriate.

- On p6, l154, in Table 1: What does “Lexical Age” refer to?

- On p7, l162: the authors suggest that they take account of a possible effect of reading experience by comparing subsamples of children matched on reading age. It would be important to explain how this matching process was done. I gather from Table 2 that “reading age” refers to “lexical age”. Is that correct? That should be stated more explicitly. Furthermore, I think the term “reading experience” is a bit unfortunate as it usually refers to the amount of experience children have reading, that is the number of books they have read. Often typically developing children will read a lot more (also at a younger age) than children with dyslexia. In the present context, it seems more appropriate to refer to a match based on “reading age” (as is done on p9, l206) or “reading skill”.

- On p7, l164: “larger” compared to what?

- On p7, l165-166: I don’t understand what the authors mean by “The latter was only significant for …”

Results

- On p8, Figure 3 and on p9, Figure 4: It would be informative if error bars could be added.

- On p8, l182-183: What do the numbers (65 ms for CTL vs. 50 ms for DYS) refer to? Where only two measures compared statistically? Why was this done with an F-test?

Discussion

- On p12, in Note1: What do the authors mean by “with strictly constant formant frequencies”?

- On p13, l323-332: I wonder whether this part can be left out as it does not seem to be directly relevant to the interpretation of the results of the present study and partly repeats what is already said.

- On p14, l335: “people” should be “children”

- On p14, l340-341: “are modulated by different other factors” is a bit vague. It would be important to be more explicit here.

English:

- On p2, l48: “contrast” seems a bit of an odd choice of word. Do the authors mean “Given the wide individual variation in manifestations of dyslexia…”?

- On p2, l51: “deficit” seem more appropriate than “component”.

- On p2, l53: “While it makes no doubt…” should be “While there is no doubt…”

- On p3, l54: “…a further question…” should be something like “…an important aim…”

- On p3, l72: “…long-standing…” should be “…protracted…”

- On p3, l79: “the age of one year old.” should be “the age of one year.”

- On p4, l91-92: “it is lesser than the one evidenced in” should be something like “to a smaller extent than”

- On p4, l94: What do the authors mean by “supplant”?

- On p4, l110: What do the authors mean by “a vocalic embryo”?

- On p5, l112: “perceive such vowel” should be “perceive such a vowel”

- On p5, l115-116: This part of the sentence is a bit hard to follow. I think the authors mean: “…was progressively reduced, eventually giving rise to the perception of ..“

- On p5, l116: “Such vocalic segment” should probably be “This vocalic segment”?

- On p10, l227: “similitudes” should probably be “similarities”?

- On p10, l239-240: “normal reading ones” would be more appropriately called “typically reading children”.

- On p11, l256: “replicated” would be more appropriate than “reduplicated”.

- On p11, l257: “VDYS” should probably “DYS”?

- On p12, l299: “those that also” should be “those who also”

- On p12, l303: “recent literature is about the” should be “recent literature concerns the”

- On p12, l306: “difficulties to capture” should be “difficulties in capturing”

- On p13, l315: “consequence” should be “consequences” or else the later verbs (“depend”) and “vary” (l316) should be changed to singular.

Author Response

ANSWERS TO THE REVIEWER are inserted below

We also added an argument in support of the relationship between allophonic perception and reading skills:

In a study with English-speaking school-age children, brain event-related potentials to a Finnish phoneme contrast were better correlated to reading skills than those with an English phoneme contrast, the higher the correlation with the foreign contrast the poorer the reading skills [31].

31.   Hämäläinen, J.; Landi, N.; Loberg, O.; Lohvansuu, K. ; Pugh, K.; Leppänen, P.H.T. Brain event-related potentials to phoneme contrasts and their correlation to reading skills in school-age children. Int. J. Behav. Devpt. 2017,

             https://doi.org/10.1177/0165025417728582  DOI: 10.1177/0165025417728582

OpenReview

English language and style

( ) Extensive editing of English language and style required
(x) Moderate English changes required
( ) English language and style are fine/minor spell check required
( ) I don't feel qualified to judge about the English language and style

Yes

Can be improved

Must be   improved

Not   applicable

Does the introduction provide sufficient background and include all   relevant references?

( )

( )

(x)

( )

Is the research design appropriate?

(x)

( )

( )

( )

Are the methods adequately described?

( )

( )

(x)

( )

Are the results clearly presented?

( )

(x)

( )

( )

Are the conclusions supported by the results?

(x)

( )

( )

( )

Comments and Suggestions for Authors

Summary

Thank you for the opportunity to review the manuscript “Enhanced sensitivity to subphonemic segments in dyslexia: a new instance of allophonic perception”. This manuscript describes an interesting experiment about subtle differences in speech perception between French typically reading children and adults, and children with dyslexia. As a consequence of co-articulation (where articulation of one sound is influenced by its surrounding sounds), the /o/ in /paRol/ (French for ‘speaking’) is influenced by the /R/ and the /l/. Taking the ‘allophonic mode of speech perception’ theory as a starting point, the authors expected that participants with dyslexia might not perceive the parts of the vowel that includes transitions from the /R/ and the /l/ as belonging to the vowel, but as separate vowels. This would lead to a different location for the phonemic boundary between /paRol/ and /paRl/ for participants with dyslexia. That was indeed what was found.

Broad comments

This is an interesting study, using a clever experimental manipulation to test the ‘allophonic mode of speech perception’ in a new way. The design seems sound and the interpretation of the results seems warranted, but I do a several broader and a rather long list of more specific comments.

- I found the manuscript a bit hard to follow, because of the large number of terms used to refer to phonological and allophonic aspects of speech perception, without explaining the differences between these terms. What for instance do the authors mean by phonological features and phonemic segments, and what are phonemic activations? Similarly, a clearer definition of allophonic units vs. allophonic features vs. allophonic segments vs. allophonic categories vs. allophonic activations should be given. And what are subphonemes (p12, l290)?

ANSWER “subphonemes” is the term chosen by Giraud & Ramus (2013 [30]) to designate allophonic segments and using this term in the present paper avoids some repetitions of “allophonic”.

ADDED (i.e. allophonic segments)

- A more detailed explanation of the relation between the ‘allophonic theory’ and the experimental design early on would make it easier to follow the rest of the paper. In the discussion (p11, l272-282) the authors provide such an explanation. I’d suggest moving that section to the introduction.

WE TRIED TO BETTER EXPLAIN THE DESIGN BY: (1) SIMPLIFYING THE PRESENTATION OF ALLOPHONES IN THE INTRODUCTION; (2) MOVING FIGURE 2 FROM THE METHOD SECTION TO THE INTRODUCTION TO SUPPORT THE PRESENTATION OF ALLOPHONES (3) MOVING THE EXPLANATION OF THE DESIGN  FROM THE DISCUSSION TO THE INTRODUCTION.

ADDED Introduction lines 83-109

“Phonological representations are the end-product of a proacted developmental process. At the start, before some six months of age, the child is endowed with universal features that do not depend on language [11]. Features are differential units and correspond to qualitative changes in the value of some acoustic property (e.g. a change in the direction of a frequency transition) [12]. The universal features are “allophonic” in the sense that they can give rise to phonological features in some languages. Somewhat later, before one year of age, allophonic features are selected and combined to constitute “phonological” features that fit into the categories of the native language [11;13]. Phonological features are in turn combined into phonemic segments, a process that ends much later, not before the ages of five and six years old [6;14].

Phoneme perception would be fairly simple if the features corresponding to the same phoneme contrast were synchronized in the speech signal. However, features are dispersed over time in the signal, giving rise to acoustic segments without definite phoneme identity [15]. Such segments constitute potential phoneme segments in other languages and are “allophonic” by definition, i.e. segments that are contextual variants of phonemes but do not pay independent contributions for distinguish words in a given language (Note [1]).

For example, in the French word /paRol/ (meaning ‘speech’), the segregation between /o/ and the surrounding /R/ and /l/ phonemes is based on changes in the frequencies of three different formants (F1, F2 and F3 “transitions”; Figure 2). However, these different acoustic changes are not synchronized and they do not coincide with the perceptual limits between /o/ and the surrounding phonemes. Such discrepancies give rise to different vocalic segments, a prototypical /o/ surrounded by /Ro/ and /ol/ transitional segments. These two latter segments are acoustically different from the prototypical /o/ and correspond to vowels that might constitute separate phonemes in some other languages.”

ADDED IN THE INTRODUCTION (TRANSPOSED FROM THE DISCUSSION): Lines 158-169 “Following allophonic theory, the paRol/paRl boundary  (i.e. the 50% response point on the durational continuum) should be located at a shorter /o/ duration children with dyslexia compared to typical-reading children because they attribute a separate identity to the transitional segments between phonemes, e.g. between /R/ and /l/. The /Rol/ segment that was temporally reduced to create the paRol/paRl continuum is composed of a prototypical /o/ (in its central portion) surrounded by /Ro/ and /ol/ transitional segments. These two latter segments are acoustically different from the prototypical/o/ and correspond to vowels that might constitute separate phonemes in some other languages. Allophonic theory postulates that people with dyslexia perceive such ‘allophones’ as separate vowel units, not as transitions between /R/ and /l/. Accordingly, people with dyslexia should rely on these vowel allophones to perceive a vowel inside the /Rl/ cluster at relatively short /Rol/ durations.

Figure 2.  Formant frequencies (PraatÓ [17]) in the word /paRol/ produced by a French speaker. The vertical lines correspond to the perceptual limits of the /Rol/ segment.

SECTION LEFT/MODIFIED IN THE DISCUSSION: lines 329-333

“Remember that following allophonic theory, the paRol/paRl boundary should be located at a shorter /o/ duration for people with dyslexia because they attribute a separate identity to the transitional segments between /R/ and /l/, and that they should rely on these allophonic segments to perceive a vowel inside the /Rl/ cluster at relatively short /Rol/ durations, a prediction that was verified in the present results.

- An more detailed explanation of the speech perception task (e.g., how many trials were included for each step along the continuum?, where trials randomized?, how long was the task?) and the procedure (where were the participants tested, how long did it take?, how was made sure that the task was engaging for children?) is essential in order for the reader to fully appreciate the experiment.

WE APOLOGIZE FOR SUCH OMISSION

ADDED lines 184-190

“Task. Each of the nine stimuli was presented 10 times for identification. The stimuli were presented in a random order with a 1000 ms intertest interval. A MatlabÓ program was used to deliver the stimuli and collect the responses. The stimuli were delivered by Samson Professional StudioÓ headphones in a quiet room. The responses were given by pressing a dedicated key on the keyboard as rapidly as possible. The participants were first familiarized with the task using the continuum endpoints. The same procedure was used for the children and the adults. The whole procedure took about 15 to 20 minutes.

- On p11, l262+l263 (+elsewhere): “people/adults/children with dyslexia” would be more appropriate than “dyslexics”.

 MODIFIED

Specific comments

Introduction

- In the first line of the introduction (p1, l28) the authors state that “Dyslexia is a specific deficit that renders an individual unable to learn to read…”. That seems a bit too drastic as most individuals with dyslexia do learn to read, but reading remains inaccurate, slow and effortful. A more nuanced definition of dyslexia would be appropriate.

MODIFIED: line 29 “ …unable to acquire fluent reading skills…”

- On p2, l35, after “dyslexia.” a reference should be included.

ADDED

- On p2, l35-36: the statement “Each deficit can in turn be expressed in different ways depending on its impact on perception, memory and attention.” is a little vague. At the very least references should be included to make clearer what the authors mean.

MODIFIED: lines 36-39 “ … as it can possibly impact different cognitive dimensions such as perception, memory and attention. These dimensions are interrelated [2] but each of them might be more or less severely affected in individuals with dyslexia depending on genetic and environmental factors.”

2.     Jones, M.R. Time, our lost dimension: Toward a new theory of perception,   attention, and memory. Psychol. Rev. 1976, 83, 323-355.

- On p2, l36-37: it is unclear what the authors mean by “interact in various different ways”. Clarify, or if the next sentence is meant as clarification, please make that connection more explicit.

MODIFIED: lines 40-41 Further, the exposure to written language will probably blur the distinction between the visual and auditory deficits.”

- On p2, l38: “audiological” should probably be “auditory”.

CORRECTED

- On p2, l44, after “writing system.” a reference should be included.

 We expressed an opinion that was the logical consequence of the previous arguments.

MODIFIED: lines 46-47 “On these grounds, individuation of the dyslexic profiles can be seen as the product of a highly specific substrate with an unsuccessful experience with the writing system”.

Materials and Methods

- On p5, l128: after “Soundforge” a reference should be included.

DONE

- On p6, l143: I don’t know what “orthophonist practitioners” are.

MODIFIED:  “speech therapists

- On p6, l144: The authors mention that the children had no oral language disorder. It should be explained on the basis of which assessment that was concluded.

ADDED : (as reported by their therapists)”

- On p6, l145: adding a brief explanation of the “Alouette Reading Test” would be appropriate.

ADDEDline 195 “ …which is based on the precision and speed in the reading of a short text (279 words).”

- On p6, l154, in Table 1: What does “Lexical Age” refer to?

REPLACED by “Reading Age”

- On p7, l162: the authors suggest that they take

account of a possible effect of reading experience by comparing subsamples of children matched on reading age. It would be important to explain how this matching process was done. I gather from Table 2 that “reading age” refers to “lexical age”. Is that correct? That should be stated more explicitly. Furthermore, I think the term “reading experience” is a bit unfortunate as it usually refers to the amount of experience children have reading, that is the number of books they have read. Often typically developing children will read a lot more (also at a younger age) than children with dyslexia. In the present context, it seems more appropriate to refer to a match based on “reading age” (as is done on p9, l206) or “reading skill”.

Reading experience was unfortunate and it was REPLACED by “Reading Age”

- On p7, l164: “larger” compared to what?

ADDED“ …than the CTL.”

- On p7, l165-166: I don’t understand what the authors mean by “The latter was only significant for …”

MODIFIED: “The reading delay was only significantly different from zero…”

 Results

- On p8, Figure 3 and on p9, Figure 4: It would be informative if error bars could be added.

ANSWER: WE DID NOT ADD error-bars because S-Deviation bars were too large, blurring the picture, and S-Error bars were too small, almost not visible.

- On p8, l182-183: What do the numbers (65 ms for CTL vs. 50 ms for DYS) refer to? Where only two measures compared statistically? Why was this done with an F-test?

ANSWER: These numbers correspond to the location of the boundary.

MODIFIED; (65 ms location; SD=5.1 ms) …

ANSWER:  A F(1,69)  test is exactly equivalent to a t(69) Student test  (F(1,df) = t² (df)).

Discussion

- On p12, in Note1: What do the authors mean by “with strictly constant formant frequencies”?

ANSWER: “strictly” was unnecessary

MODIFIED & ADDED p.14, Note 1: “ …temporal reduction of a /o/ segment with constant formant frequencies (i.e. the /Rol/ segment was replaced by a segment generated by reduplication of a central period at the middle of the segment).

- On p13, l323-332: I wonder whether this part can be left out as it does not seem to be directly relevant to the interpretation of the results of the present study and partly repeats what is already said.

ANSWER: We agree and dropped this part in the Discussion but we added the last portion in the Introduction. (The portion staring with: “The question then is why should such competition still raise severe problems for learning to read? …)

- On p14, l335: “people” should be “children”

MODIFIED

- On p14, l340-341: “are modulated by different other factors” is a bit vague. It would be important to be more explicit here.

ADDED at the end of the Discussion, lines 381-391;

“Timing constraints might impact the synchronization of grapheme and phoneme decoding in the temporal cortex, which is vital for learning to read [23]. Phoneme decoding is slower than allophonic decoding in children with dyslexia [20], a difference that might be larger in children with a RAN deficit. The temporal delay between phonemic and allophonic decoding might also vary with individual differences in the access to phonological representations in the prefrontal cortex, such differences being correlated with phonemic awareness [9]. A higher degree of phonemic awareness might correspond to a quicker access to the prefrontal cortex and, in turn, a faster transmission of the phonemic codes to the temporal cortex. Finally, it should be stressed that is not only the access to the prefrontal cortex but also the nature of the prefrontal representations (phonemic vs. allophonic: [36]) that should affect the transmission of the phonemic codes to the temporal cortex.

 “

MODIFIED in the Conclusions, lines 399-400

“The implications of allophonic perception for reading acquisition are modulated by factors such as RAN and phonemic awareness skills that might reflect individual differences in the synchronization of grapheme and phoneme decoding.

English:

- On p2, l48: “contrast” seems a bit of an odd choice of word. Do the authors mean “Given the wide individual variation in manifestations of dyslexia…”?

MODIFIED as suggested

- On p2, l51: “deficit” seem more appropriate than “component”.

MODIFIED

- On p2, l53: “While it makes no doubt…” should be “While there is no doubt…”

MODIFIED

- On p3, l54: “…a further question…” should be something like “…an important aim…”

MODIFIED

- On p3, l72: “…long-standing…” should be “…protracted…”

MODIFIED

- On p3, l79: “the age of one year old.” should be “the age of one year.”

Passage deleted.

- On p4, l91-92: “it is lesser than the one evidenced in” should be something like “to a smaller extent than”

MODIFIED  line 128 “… though to a smaller extent than”

- On p4, l94: What do the authors mean by “supplant”?

MODIFIED: line 130  “…suggesting that these children would not use allophonic units”

- On p4, l110: What do t

he authors mean by “a vocalic embryo”?

MODIFIED line 153 “segment”

- On p5, l112: “perceive such vowel” should be “perceive such a vowel”

MODIFIED

- On p5, l115-116: This part of the sentence is a bit hard to follow. I think the authors mean: “…was progressively reduced, eventually giving rise to the perception of ..“

MODIFIED

- On p5, l116: “Such vocalic segment” should probably be “This vocalic segment”?

MODIFIED

- On p10, l227: “similitudes” should probably be “similarities”?

MODIFIED

- On p10, l239-240: “normal reading ones” would be more appropriately called “typically reading children”.

MODIFIED

- On p11, l256: “replicated” would be more appropriate than “reduplicated”.

MODIFIED

- On p11, l257: “VDYS” should probably “DYS”?

MODIFIED

- On p12, l299: “those that also” should be “those who also”

MODIFIED

- On p12, l303: “recent literature is about the” should be “recent literature concerns the”

MODIFIED

- On p12, l306: “difficulties to capture” should be “difficulties in capturing”

MODIFIED

- On p13, l315: “consequence” should be “consequences” or else the later verbs (“depend”) and “vary” (l316) should be changed to singular.

MODIFIED “consequences”

Note [1]. This definition of the allophone is similar to the one proposed by Trask (1996, p.14); [16]: “One of two or more phonetically distinct segments which can realize a single phoneme in varying circumstances.”

Round  2

Reviewer 1 Report

The authors have addressed all my concerns. In particular, they have expanded the introduction section to discuss the evidence that deficits in phonological awareness may not stem from deficits in categorical perception, and they have expanded the discussion section to clarify the interpretation of the findings. One of my concerns was the fact that this study included a group of adults to account for possible effects of chronological age instead of the more common option of including a group of controls matched by chronological age. I think that this is still a limitation of the present design, but the authors have now acknowledged this fully in the discussion, and this is now reflected in their interpretation of the comparisons between child and adult participants' performance. 

The revised version of the manuscript has several typos, so the manuscript would benefit from a more thorough check and editing. 

Reviewer 2 Report

The authors have addressed the concerns I raised in my previous review in a comprehensive manner. I think the paper in its present form (after some minor text editing) will make an interesting addition to the literature.